# Detection rate and treatment gap for atrial fibrillation identified through screening in community health centers in China (AF-CATCH): A prospective multicenter study

Yi Chen[1], Qi-Fang Huang[1], Chang-Sheng Sheng[1], Wei Zhang[1], Shuai Shao[1], Dian Wang[1], Yi-Bang Cheng[1], Ying Wang[1], Qian-Hui Guo[1], Dong-Yan Zhang[1], Yan Li[1], Nicole Lowres[2], Ben Freedman[2], Ji-Guang Wang[1]*

1 Center for Epidemiological Studies and Clinical Trials, Shanghai Key Laboratory of Hypertension, The Shanghai Institute of Hypertension, Department of Hypertension, Ruijin Hospital, Shanghai Jiao Tong University School of Medicine, Shanghai, China, 2 Heart Research Institute, Sydney Medical School, Charles Perkins Center, and Cardiology Department, Concord Hospital, The University of Sydney, Sydney, Australia

* jiguangwang@aim.com

**Data Availability Statement:** Data cannot be shared publicly because of ethical restrictions. Data are available from Ruijin Hospital Ethics Committee

## Abstract

### Background

Atrial fibrillation (AF) is underdiagnosed and especially undertreated in China. We aimed to investigate the prevalence of unknown and untreated AF in community residents ($\geq$65 years old) and to determine whether an education intervention could improve oral anticoagulant (OAC) prescription.

### Methods and findings

We performed a single–time point screening for AF with a handheld single-lead electrocardiography (ECG) in Chinese residents ($\geq$65 years old) in 5 community health centers in Shanghai from April to September 2017. Disease education and advice on referral to specialist clinics for OAC treatment were provided to all patients with actionable AF (newly detected or undertreated known AF) at the time of screening, and education was reinforced at 1 month. Follow-up occurred at 12 months. In total, 4,531 participants were screened (response rate 94.7%, mean age 71.6 ± 6.3 years, 44% male). Overall AF prevalence was 4.0% (known AF 3.5% [$n = 161$], new AF 0.5% [$n = 22$]). The 183 patients with AF were older ($p < 0.001$), taller ($p = 0.02$), and more likely to be male ($p = 0.01$), and they had a higher prevalence of cardiovascular disease than those without AF ($p < 0.001$). In total, 85% (155/183) of patients were recommended for OAC treatment by the established guidelines (CHA$_2$DS$_2$-VASc $\geq$ 2 for men; $\geq$ 3 for women). OAC prescription rate for known AF was 20% (28/138), and actionable AF constituted 2.8% of all those screened. At the 12-month follow-up in 103 patients (81% complete), despite disease education and advice on specialist referral, only 17 attended specialist clinics, and 4 were prescribed OAC. Of those not attending specialist clinics, 71 chose instead to attend community health centers or secondary hospital clinics, with none prescribed OAC, and 15 had no review. Of the 17 patients

(contact via wyfkjc@163.com) for researchers who meet the criteria for access to confidential data.

**Funding:** J-GW is financially supported by grants from the National Natural Science Foundation of China (81400346, 81470533, 91639203, 81770418, and 81770455), Ministry of Science and Technology (grant 2015AA020105-06), and Ministry of Health (grants 2016YFC0900902), Beijing, China, and Science and Technology Commissions (grant 15XD1503200), and Population and Family Planning Commissions of Shanghai Municipal (grants 15GWZK0802 and 2017BR025), Shanghai, China. NL is funded by an NSW Health Early Career Fellowship (H16/ 52168). No funding bodies had any role in study design, data collection and analysis, decision to publish, or preparation of the manuscript.

**Competing interests:** I have read the journal's policy and the authors of this manuscript have the following competing interests: BF reports grants, personal fees, and non-financial support from Bayer; grants, personal fees, and non-financial support from BMS-PFizer; personal fees and non-financial support from Daiichi-Sankyo, outside the submitted work; and personal fees and non-financial support by Omron. The remaining authors have declared that no competing interests exist. AliveCor provided ECG Heart Monitors for study purposes: the investigators are not affiliated with or have any financial or other interest in AliveCor.

**Abbreviations:** AF, atrial fibrillation; CI, confidence interval; ECG, electrocardiography; ESC, European Society of Cardiology; ICH, intracerebral hemorrhage; INR, international normalized ratio; NOAC, new oral anticoagulant; OAC, oral anticoagulant; REHEARSE-AF, Remote Heart Rhythm Sampling Using the AliveCor Heart Monitor to Screen for Atrial Fibrillation; SD, standard deviation; STROBE, Strengthening the Reporting of Observational Studies in Epidemiology; TIA, transient ischemic attack.

with new AF and a class 1 recommendation for OAC, only 3 attended a specialist clinic, and none were prescribed OAC. Of the 28 AF patients taking OAC at baseline, OAC was no longer taken in 4. Ischemic stroke ($n = 2$) or death ($n = 3$) occurred in 5/126 (4%), with none receiving OAC. As screening was performed at a single time point, some paroxysmal AF cases may have been missed; thus, the rate of new AF may be underestimated.

## Conclusions

We demonstrated a noticeable gap in AF detection and treatment in community-based elderly Chinese: actionable AF constituted a high proportion of those screened. Disease education and advice on specialist referral are insufficient to close the gap. Before more frequent or intensive screening for unknown AF could be recommended in China, greater efforts must be made to increase appropriate OAC therapy in known AF to prevent AF-related stroke.

## Author summary

### Why was this study done?

- Atrial fibrillation (AF) is a common heart rhythm problem that often has no symptoms, so it is often underdiagnosed.

- People with AF can have a very high stroke risk, which is highly preventable with appropriate oral anticoagulant (OAC) medications.

- Neither the prevalence of unknown and untreated AF in the Chinese community nor whether patient education in the community health center has the potential to improve OAC prescription are known.

### What did the researchers do and find?

- We screened for AF in residents aged ≥65 years in community health centers in Shanghai and provided disease education and advice on referral to specialist clinics for OAC treatment to people with newly detected or undertreated known AF.

- We demonstrated a noticeable gap in AF detection and treatment: 2.8% of those screened had unknown or untreated AF.

- At 12 months, only 17/103 people with newly detected or undertreated known AF attended specialist clinics, and only 4/17 had commenced OAC therapy.

### What do these findings mean?

- We highlight a serious public health issue in China with underdiagnosis and undertreatment of AF in the community that requires a whole-of-system approach.

- To prevent AF-related stroke in China, greater efforts must be made to increase appropriate OAC therapy in people with AF.

## Introduction

Atrial fibrillation (AF) is a growing problem in cardiovascular disease, with age-adjusted incidence rates on the rise [1]. It is predicted that AF prevalence will at least double in the next 30 years [2,3]. Patients with AF have about 5-fold increased risk of ischemic stroke [4,5], which is highly preventable with appropriate oral anticoagulant (OAC) therapy [6]. However, AF may be asymptomatic and unrecognized prior to stroke. Approximately 10% of ischemic strokes are caused by AF that is first detected at the time of stroke [7]. A systematic review showed that unknown asymptomatic AF was common, occurring in 1.4% of those aged 65 years or older on a single–time point check for presence of AF [8], which is also confirmed in a more recent individual patient meta-analysis [9]. It is therefore intuitive that population-based screening for asymptomatic AF and subsequent anticoagulant treatment may be a promising public health strategy to prevent stroke [10].

In China, AF is underdiagnosed and especially undertreated. A multicenter study in China showed that 7.9% of the patients with ischemic stroke and transient ischemic attack (TIA) had newly detected AF, with 3.5% detected by electrocardiography (ECG) and 4.4% detected by 6-day Holter monitoring [11]. The China National Stroke Screening Survey (community-based) data showed that only 2.2% of patients with ischemic stroke and AF were taking OAC at the time of stroke [12]. Our previous study in people aged ≥65 years in Shanghai showed that 88.6% of patients with AF detected by a single–time point ECG were unaware of their disease, and only 1% of these patients were on anticoagulant therapy [13]. According to a recent survey, low OAC rates were related to patients' unawareness that they had AF (26%), lack of AF symptoms (35%), and a lack of understanding of the risks associated with AF (22%) [14].

This study aimed to investigate the prevalence of unknown and untreated AF in residents aged ≥65 years in urban Shanghai and investigate an educational intervention to improve OAC prescription in patients with "actionable AF" (i.e., those identified as newly diagnosed AF or undertreated known AF who have a class 1 recommendation for OAC thromboprophylaxis according to the 2016 European Society of Cardiology (ESC) guidelines (i.e., $CHA_2DS_2$-VASc score $\geq 2$ for men or $\geq 3$ for women)) [5].

## Methods

This study was a prospective cross-sectional study conducted in 5 communities (Yuyuan Community, Laoximen Community, Ruijin Second Road Community, Puxing Community, Sanlin Community) in urban areas of Shanghai from April to September 2017. This study was part of a larger study (AF-CATCH, CT02990741), and the protocol has been previously published in detail [15]. The study was approved by the Ethics Committee of Ruijin Hospital, Shanghai Jiaotong University School of Medicine. Our study was conducted in accordance with the principles of the Declaration of Helsinki. This study is reported as per the Strengthening the Reporting of Observational Studies in Epidemiology (STROBE) guideline (S1 STROBE Checklist).

## Study population

Our study subjects were residents aged 65 years or older recruited from 5 community health centers (Yuyuan Community Health Center, Laoximen Community Health Center, Ruijin Second Road Community Health Center, Puxing Community Health Center, Sanlin Community Health Center) in urban areas of Shanghai from April to September 2017. A description of the community health centers in China is provided in the S1 Text. The screening program was publicized through a public health press conference and media release in Shanghai, official notices of the neighborhood committee, and placement of posters in community health centers. All residents aged 65 years or older were eligible for participation. All participants were informed of the study design and gave their written informed consent before joining the screening program. The consent included agreements to share their information for confidential academic analysis and agreements with the follow-up arrangements in the study.

## Screening and intervention

In a screening clinic visit at the community health center, a single-lead (lead I) ECG was recorded for 30 seconds with a handheld ECG device (AliveCor Heart Monitor, now Kardia Mobile). Each ECG rhythm strip was reviewed by a cardiologist from the research team (YC) at the screening visit. The ECGs were classified into 3 groups: sinus rhythm, AF, and uninterpretable. Participants with uninterpretable single-lead ECG were referred for 12-lead ECGs, which were reviewed by a second cardiologist (DW or J-GW). Both AF and atrial flutter diagnosed by ECG were identified as cases of AF. A questionnaire regarding medical history, lifestyle, and use of medications was administered to all participants by the research cardiologists (S2 Text). Participants with a documented history of AF in their medical records from qualified hospitals or with AF recorded on any prior ECG, who were in sinus rhythm on the screening ECG, were defined as "known AF in sinus rhythm." ECGs and medical records obtained outside the study center were documented for verification. Those without AF history or AF rhythm were candidates for an ongoing trial that aims to determine the incidence rate of unknown AF at single–time point screening and during more intensive subsequent screening (AF-CATCH, NCT02990741) [15].

For patients with AF history or AF rhythm, $CHA_2DS_2$-VASc score was calculated to estimate the risk of stroke and determine eligibility for OAC according to the 2016 ESC guidelines (i.e., $CHA_2DS_2$-VASc $\geq$ 2 for men or $\geq$ 3 for women: class 1 OAC recommendation) [5]. Details of current antithrombotic treatment were confirmed and recorded using the patients' medical record (S3 Text). All participants not receiving guideline-recommended OAC were deemed actionable AF (including both known AF and newly detected AF) and were included in our educational intervention program. The education program included one-on-one disease education with the research cardiologist at the community health center and provision of educational materials. Education involved communicating that AF is associated with high risk of stroke, the appropriate treatment options, and benefits and risks of anticoagulants. Because warfarin, non–vitamin K OACs, and international normalized ratio (INR) testing are not available at community health centers, patients were advised to attend a specialist clinic for review and OAC prescription. Information on how to make an appointment at a designated specialist clinic near each community was also provided. Across Shanghai, there are over 20 tertiary hospitals with specialist cardiovascular clinics, 10 of which also have a specialist AF clinic. These tertiary hospitals are located between 2 and 10 km from the community health centers and provide the majority of OAC prescription and INR testing for the region.

At 1 month, attendance at the specialist clinic was confirmed via telephone. If they had not attended, we invited them to return for review and another single-lead ECG at the community health center. Education and advice on specialist referral were reinforced.

### Follow-up

Follow-up occurred at 12 months in the community health centers. A follow-up questionnaire was administered by the research cardiologist regarding living status; adverse outcomes, including stroke and myocardial infarction, in the past 1 year; whether they attended the AF specialist clinic or not; what treatment they received from specialists, especially in relation to OAC prescription; and, if relevant, why they did not attend the AF specialist clinic (S4 Text). If participants were unable to attend a visit in the community health center, telephone follow-up was offered. If patients had not attended a specialist clinic within the 12-month period, AF education and the importance of attending a designated AF specialist clinic were reinforced.

### Statistical analysis

Statistical analyses were performed in accordance with our statistical analysis plan using SAS 9.3 (SAS Institute Inc., Cary, NC, United States) [15]. Continuous variables were presented as means with standard deviation (SD), and categorical variables were presented as percentages. New episodes of AF were expressed as true positives divided by the total number screened with accompanying binomial 95% confidence intervals (CIs) calculated using Clopper–Pearson methodology. To assess baseline characteristic differences between the AF patients and non-AF participants, the $\chi^2$ test or Fisher's exact test was used to compare categorical variables, and Student $t$ test was used to compare continuous variables. For all analyses, a two-sided $p$-value $< 0.05$ was considered statistically significant.

## Results

### Characteristics of the study population

It is estimated that the 5 participating community health clinics service a total of 9,710 people aged 65 years and over each year. During the study period, 4,784 residents attended the clinic and were approached to participate in the study, and 253 declined (response rate 94.7%). A total of 4,531 citizens (2,530 women [55.8%]; mean [± SD] age 71.6 ± 6.3 years) participated in the screening program. The prevalence of AF was 4.0% (95% CI 3.5%–4.7%) ($n = 183$), including 1.8% (95% CI 1.4%–2.2%) with ECG-confirmed known AF ($n = 82$), 1.7% (95% CI 1.4%–2.2%) with known AF in sinus rhythm ($n = 79$), and 0.5% (95% CI 0.3%–0.7%) with ECG-confirmed previously unknown AF ($n = 22$). In men as well as women, the prevalence of AF was higher with advancing age ($p$ for trend $< 0.001$ for both sexes, Fig 1). The 183 patients with AF were older, taller, and more likely to be male and had a higher prevalence of prior stroke or TIA, coronary heart disease, and congestive heart failure ($p < 0.02$, Table 1). The heart rate of those with AF was higher ($p < 0.001$). Body mass index, blood pressure, prevalence of smoking, alcohol intake, hypertension, and diabetes mellitus or use of antihypertensive drugs did not differ between the 2 groups. Patients with new AF were older than those with known AF (77.5 ± 7.8 years versus 74.7 ± 7.7 years, $p = 0.07$) and had lower $CHA_2DS_2$-VASc scores (3.0 ± 1.3 versus 3.5 ± 1.6, $p = 0.08$), though the differences were not statistically significant. The majority of new AF cases (18/22) had no symptoms, whereas 4/22 complained of shortness of breath and dizziness.

### Management of AF at baseline

The majority of the patients with AF (155/183, 85%) had a class 1 recommendation for OAC therapy [5]: specifically, 138/161 (86%) known AF and 17/22 (77%) new AF (Fig 2). Of the OAC-eligible patients, 11.0% (17/155) had a HAS-BLED score $\geq 3$, and 4 patients with known AF had previously stopped warfarin because of bleeding (2 retinal hemorrhage, 2 hematuria).

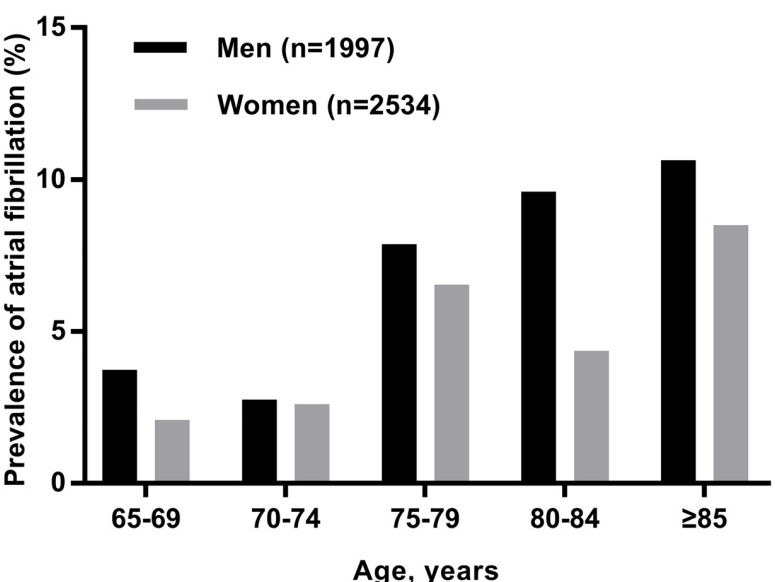

**Fig 1. Prevalence of atrial fibrillation according to age and sex.**

OAC prescription rate (for those with a class 1 recommendation) was 20% (28/138) in known AF patients and 0% in new patients with AF (Fig 2). Of the 28 patients on OAC therapy (24 warfarin, 4 dabigatran), 5 patients had been prescribed warfarin after valve implantation. Therefore, OAC was taken by only 17% of patients with AF (23/133) without implanted heart valves (22/70 patients in AF rhythm [31%] versus 1/63 in sinus rhythm [2%, $p < 0.001$]). In those who were taking warfarin, the majority (14/17, 82%) had recent INR values between 2.0 and 3.0. Thirty patients (27.5%) were on antiplatelet therapy only: aspirin (15%), clopidogrel

**Table 1. Characteristics of the study population.**

| Characteristic | AF (*n* = 183) | Non-AF (*n* = 4,348) | *p* |
|---|---|---|---|
| Women, *n* (%) | 86 (47.0) | 2,444 (56.3) | 0.01 |
| Age, years | 74.8 ± 7.3 | 71.4 ± 6.3 | <0.001 |
| Body height, cm | 161.3 ± 9.8 | 159.7 ± 8.8 | 0.02 |
| Body mass index, kg/m$^2$ | 24.9 ± 4.2 | 24.6 ± 3.5 | 0.30 |
| Systolic blood pressure, mm Hg | 134.5 ± 18.4 | 136.6 ± 18.8 | 0.13 |
| Diastolic blood pressure, mm Hg | 73.6 ± 11.0 | 73.6 ± 9.4 | 0.99 |
| Heart rate, beats/min | 78.1 ± 16.5 | 73.7 ± 10.9 | <0.001 |
| Current smoking, *n* (%) | 17 (9.4) | 596 (13.8) | 0.10 |
| Alcohol intake, *n* (%) | 24 (13.3) | 493 (11.4) | 0.43 |
| Hypertension, *n* (%) | 114 (62.3) | 2,444 (56.2) | 0.10 |
| Use of antihypertensive drugs, *n* (%) | 101 (55.2) | 2,277 (52.4) | 0.13 |
| Diabetes mellitus, *n* (%) | 50 (27.3) | 1,007 (23.2) | 0.20 |
| History of cardiovascular disease | | | |
| Coronary heart disease, *n* (%) | 25 (13.6) | 271 (6.2) | <0.001 |
| Congestive heart failure, *n* (%) | 5 (2.7) | 18 (0.4) | <0.001 |
| Stroke or TIA, *n* (%) | 45 (24.6) | 704 (16.6) | 0.004 |

Values are means ± standard deviation or number of participants (% of column total).

Abbreviations: AF, atrial fibrillation; TIA, transient ischemic attack.

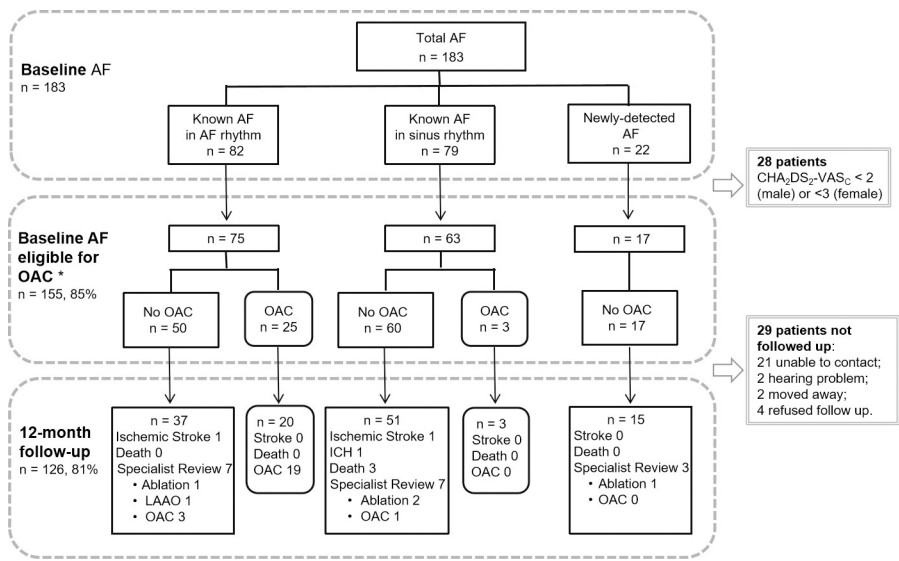

**Fig 2. The flow chart of patients with AF.** *Patients with AF of $CHA_2DS_2$-VASc $\geq$ 2 (men) or $\geq$ 3 (women). AF, atrial fibrillation; LAAO, left atrial appendage occlusion; OAC, oral anticoagulant; ICH, intracerebral hemorrhage.

(9%), and aspirin plus clopidogrel (4%). Antiarrhythmic agents were more often used in known AF in sinus rhythm (25% versus 7%), whereas rate control medications were more common for those with known AF in AF rhythm (60% versus 30%, Table 2).

**Table 2. Management of AF in patients eligible for oral anticoagulant therapy* ($n$ = 155).**

| Characteristic | Known AF in AF rhythm | Known AF in sinus rhythm | Previously unknown AF |
|---|---|---|---|
| | $n$ = 75 | $n$ = 63 | $n$ = 17 |
| Men, $n$ (%) | 37 (49) | 32 (51) | 11 (65) |
| Age, years | 75.6 ± 7.7 | 74.7 ± 7.9 | 80.2 ± 6.4 |
| AF duration, years | 10.1 ± 10.7 | 5.6 ± 5.7 | 0 |
| $CHA_2DS_2$-VAScscore | 4.0 ± 1.5 | 4.0 ± 1.5 | 3.5 ± 0.9 |
| | Baseline | Baseline | 12-month follow-up |
| $n$ (%) | $n$ = 75 | $n$ = 63 | $n$ = 15 |
| AF catheter ablation | 5 (7) | 10 (16) | 1 (7) |
| Antiarrhythmic agents§ | 5 (7) | 16 (25) | 4 (25) |
| Rate control | 45 (60) | 19 (30) | 8 (47) |
| Digoxin | 20 (27) | 4 (6) | 2 (11) |
| β-Blocker | 35 (47) | 17 (27) | 8 (42) |
| Oral anticoagulant | 25 (33) | 3 (5) | 0 |
| Warfarin | 22 (29) | 2 (3) | 0 |
| Dabigatran | 3 (4) | 1 (2) | 0 |
| Antiplatelet drugs only | 18 (24) | 20 (32) | 5 (33) |
| Aspirin | 12 (16) | 9 (14) | 3 (20) |
| Clopidogrel | 5 (7) | 7 (11) | 2 (11) |
| Dual antiplatelet | 1 (1) | 4 (6) | 0 |

*Patients with AF of $CHA_2DS_2$-VASc $\geq$ 2 (men) or $\geq$ 3 (women).

§Antiarrhythmic agents = propafenone or amiodarone.

Abbreviation: AF, atrial fibrillation

All 127 patients with actionable AF (110 undertreated AF and 17 new AF) received education at the baseline visit. Education and advice on specialist referral were reinforced at 1 month in 108 participants, either during a community health clinic visit ($n = 84$) or via telephone consultation ($n = 24$).

## Follow-up at 12 months

Follow-up occurred in 148/183 patients at 12 months (81% complete): 96 were in person at the community health center, and 52 were by telephone. From the OAC-eligible patients, 126/155 (81%) were followed up, and 29 were lost to follow-up (21 unable to contact, 4 declined a phone interview, 2 had hearing difficulty and lived alone, and 2 had moved and were not available) (Fig 2). At 12 months, 3 patients died (1 pneumonia, 1 heart failure, and 1 unknown causes), 3 had nonfatal stroke (2 ischemic and 1 intracranial hemorrhage), and 4 had experienced an acute coronary syndrome (no myocardial infarction) (Fig 2). None of these patients were taking OAC, and only one of those who died was taking antiplatelet agents (Fig 2).

In total, 103/127 patients with actionable AF were followed up at 12 months (81% complete), and only 17 attended cardiovascular specialist clinics, with 4 of these being prescribed OAC (warfarin for all 4 patients, Fig 2). Most patients ($n = 71$) did not attend a specialist clinic and instead attended community health centers or outpatient services of smaller hospitals, where they were prescribed traditional Chinese medicine ($n = 34$) or antiplatelet agents ($n = 12$) or both ($n = 3$). Fifteen patients did not go to any doctors and did not take any antithrombotic drugs. For the 5 patients who underwent either left atrial appendage occlusion ($n = 1$) or catheter ablation ($n = 4$), dabigatran was prescribed 3 months before and after procedure and then discontinued. Of the 17 patients with new AF and a class 1 recommendation for OAC, only 3 attended a specialist clinic and, none were prescribed OAC (Fig 2). Of the 28 patients with AF taking OAC at baseline, OAC was no longer taken in 4, including 3 in sinus rhythm versus 1 in AF rhythm (Fig 2).

## Discussion

The main findings of our study are that AF is prevalent in Chinese people aged 65 years and older, with an overall prevalence of 4.0%, and that actionable AF constituted 2.8% of all those screened, a very high proportion in whom initiation of OAC therapy could make a difference to prognosis. Of those with known AF qualifying for guideline-recommended OAC prophylaxis, only 20% were taking OAC. The percentage was even lower (17%) when patients with valvular prostheses requiring warfarin were excluded, highlighting the significant undertreatment of AF in patients managed by community health centers. Ischemic stroke or death occurred in 5 patients (none were taking OAC) in 1 year, which was potentially preventable if guideline-recommended OAC had been used.

Our data highlight a serious public health issue in China with undertreatment of AF in the community, which is likely to result in an excess of preventable stroke and death. Our OAC treatment rates are lower than those reported in Chinese tertiary hospitals (9.6% to 68.4%) and nontertiary hospitals (4.0% to 28.2%) [16]; however, our study may more closely reflect a real-world sample of the community-dwelling population aged ≥65 years in China. Despite providing education regarding AF and advice on specialist referral for all 127 patients with actionable AF, the majority did not attend a specialist clinic, and of the 17 who attended the specialist clinic, only 4 were commenced on OAC therapy.

Low prescription of OAC is complex and compounded by many factors, including (1) patient reluctance to attend because they did not consider themselves very ill, despite one-on-one education; (2) patients elected to attend their local community health center and believed

their AF was adequately treated, even though they were only prescribed antiplatelet drugs and traditional Chinese medicine; and (3) community health center physicians may lack knowledge regarding evidence-based management of AF. Furthermore, there was also low adherence to AF guidelines by the specialists in the tertiary hospital clinics, which may be the result of (1) worries about patient treatment compliance to warfarin in the absence of a dedicated anticoagulant clinic or team in the local area; (2) the common misperception of aspirin efficacy and safety [17,18]; (3) restricted use and/or unavailability of the new OACs (NOACs), even in some tertiary hospitals; and (4) the expense of NOACs may prohibit patient agreement to the prescription.

Education for both patients and physicians in the community health centers is required to overcome these barriers. It has been demonstrated across 5 countries that multifaceted education programs targeting both patients and providers can significantly increase the proportion of patients treated with OAC from 68% to 80% at 1 year [19]. In China, it may be possible to run workshops providing simplified important information on AF. Workshops for patients should focus on disease education to improve health-seeking behaviors, whereas physician education should focus more on understanding evidence-based treatment for AF and establishment of direct and efficient referral pathways for patients to access appropriate treatment with a cardiologist. Directed health resource allocations for patients to access anticoagulant drugs and INR testing in community health centers, such as a regular specialist outreach clinic, may help enhance adherence and long-term persistence with OAC. Furthermore, public health policies that include NOACs into medical insurance will give both patients and doctors more alternatives to choose from [20].

Designated pathways to treatment are very important if AF screening is to be undertaken. In AF screening studies, it is well documented that the success for guideline adherence to OAC treatment is influenced by the pathway to treatment offered within the study. In the Swedish STROKESTOP study, an OAC review with the study cardiologist was part of the screening process, and 99% were reviewed, resulting in 74% of people with actionable AF having OAC initiated [21]. However, the Belgium Heart Rhythm Week study, which screened in the community and had a similar pathway to our study (i.e., providing advice for patients to consult their general practitioner or cardiologist), found that only 11.2% of eligible participants with actionable AF commenced OAC therapy [22]. Recently, the Huawei Heart Study [23], using a program of integrated AF management directed by a mobile AF application in China, reported that approximately 80% of OAC-eligible patients with AF were anticoagulated. However, in the Huawei Heart Study, there is a likely selection bias because participants all owned smartphones, were able to download the app, and chose to use smart wearable devices to monitor their own pulse rhythm. The cohort was much younger and likely had higher health literacy and interest in improving their health and, therefore, were more likely to respond to physician advice.

Our finding of a detection rate of 0.5% for new AF cases is comparable to the 0.5%–0.8% identified in other mass community screening programs using a single-time screen [21,24]. However, we detected fewer cases of new AF than we anticipated compared with the result of 2.0% identified in 2011 in our previous study [13]. The lower detection rate may be related to the fact that 82% of our participants had undergone an ECG in the past 2 years as part of the free annual health examinations (including a 12-lead ECG) in community health centers for people ≥65 years old, which have been organized and supported by the Chinese government over the past 5 years. This may support the notion that annual ECG checks indeed help to detect AF early. However, annual ECG checks or single-time handheld ECG screens may still underestimate the prevalence of AF because cases of paroxysmal AF may not be captured. Innovative approaches involving current mobile and wireless technologies to record multiple

ECG snapshots with patient-activated handheld single-lead ECG devices may help improve the detection rate [25]. This was demonstrated in the STROKESTOP study, where 0.5% new AF was identified with the first screen, and after 2 weeks of intermittent twice-daily screening, the yield of new AF increased to 3% [21]. The Remote Heart Rhythm Sampling Using the AliveCor Heart Monitor to Screen for Atrial Fibrillation (REHEARSE-AF) study also identified a higher yield of 3.8% using 1–2 ECG recordings per week over 1 year [26]. However, before more frequent or intensive screening for unknown AF could be recommended in China, greater effort must be made to increase OAC treatment in known AF.

There are some limitations to our study that warrant discussion. First, the screening was performed at a single time point, and some cases of paroxysmal AF may have been missed; thus, the detection rate of AF may be underestimated. Second, despite our strict follow-up procedures, the loss to follow-up was 19.1%. Third, our study protocol did not allow the research cardiologist to directly review and treat patients with OAC; however, this reflects the real-world pathway to treatment for community-based Chinese. Finally, we did not involve physicians in the community health centers into our screening and education program to encourage initiation of and adherence with OAC, which may also contribute to the low rate of referral.

## Conclusions

There is a noticeable gap in the detection and treatment of AF in the Chinese residents aged 65 and over in urban Shanghai. AF disease education and advice on specialist referral provided in our study was insufficient to close the treatment gap. Currently, to prevent AF-related stroke in China, greater efforts must be made to increase appropriate OAC therapy in known AF, such as more effective downstream management pathways and health resource allocations, before more frequent or intensive screening for unknown AF could be recommended.

## Supporting information

**S1 STROBE Checklist. STROBE, Strengthening the Reporting of Observational Studies in Epidemiology.**
(DOCX)

**S1 Text. The health system in China.**
(DOCX)

**S2 Text. Screening questionnaire.**
(DOCX)

**S3 Text. AF questionnaire.**
(DOCX)

**S4 Text. AF follow-up questionnaire.**
(DOCX)

## Acknowledgments

The authors gratefully acknowledge the voluntary participation of all study subjects, the technical assistance of the physicians and nurses of the community health centers, and the expert assistance of Yu-Ting Jiang, Jun-Wei Li, Bei-Wen Lv, Jia-Ye Qian, Yu-Zhu Shi, Yi-Ni Zhou, Yi Zhou, and Jia-Jun Zong from the Shanghai Institute of Hypertension (Shanghai, China).

## Author Contributions

**Conceptualization:** Yi Chen, Qi-Fang Huang, Chang-Sheng Sheng, Ben Freedman, Ji-Guang Wang.

**Data curation:** Yi Chen, Qi-Fang Huang, Chang-Sheng Sheng, Wei Zhang, Ji-Guang Wang.

**Formal analysis:** Yi Chen, Nicole Lowres, Ben Freedman, Ji-Guang Wang.

**Funding acquisition:** Ji-Guang Wang.

**Investigation:** Yi Chen, Qi-Fang Huang, Chang-Sheng Sheng, Wei Zhang, Shuai Shao, Dian Wang, Yi-Bang Cheng, Ying Wang, Qian-Hui Guo, Dong-Yan Zhang.

**Methodology:** Yi Chen, Qi-Fang Huang, Chang-Sheng Sheng, Wei Zhang, Shuai Shao, Dian Wang, Yi-Bang Cheng, Ying Wang, Qian-Hui Guo, Dong-Yan Zhang, Nicole Lowres, Ben Freedman, Ji-Guang Wang.

**Project administration:** Yi Chen, Ben Freedman, Ji-Guang Wang.

**Supervision:** Yan Li, Nicole Lowres, Ben Freedman, Ji-Guang Wang.

**Validation:** Dian Wang, Yan Li, Nicole Lowres, Ben Freedman, Ji-Guang Wang.

**Visualization:** Ben Freedman.

**Writing – original draft:** Yi Chen.

**Writing – review & editing:** Yi Chen, Qi-Fang Huang, Chang-Sheng Sheng, Wei Zhang, Shuai Shao, Dian Wang, Yi-Bang Cheng, Ying Wang, Qian-Hui Guo, Dong-Yan Zhang, Yan Li, Nicole Lowres, Ben Freedman, Ji-Guang Wang.

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
