## [Decision Letter · Decision Letter 0]

24 Feb 2020

Dear Dr. Wang,

Thank you very much for submitting your manuscript "Can we close the atrial fibrillation detection and treatment gap in China by screening and education in community health centers?" (PMEDICINE-D-19-04467) for consideration at PLOS Medicine. 

[LINK]

In light of these reviews, I am afraid that we will not be able to accept the manuscript for publication in the journal in its current form, but we would like to consider a revised version that addresses the reviewers' and editors' comments. Obviously we cannot make any decision about publication until we have seen the revised manuscript and your response, and we plan to seek re-review by one or more of the reviewers. 

We expect to receive your revised manuscript by Mar 09 2020 11:59PM. Please email us (plosmedicine@plos.org) if you have any questions or concerns.

We look forward to receiving your revised manuscript. 

Sincerely,

Adya Misra, PhD

Senior Editor 

PLOS Medicine

plosmedicine.org

Title- Please revise your title according to PLOS Medicine's style. Your title must be nondeclarative and not a question. It should begin with main concept if possible. "Effect of" should be used only if causality can be inferred, i.e., for an RCT. Please place the study design ("A randomized controlled trial," "A retrospective study," "A modelling study," etc.) in the subtitle (ie, after a colon).

Abstract- please clarify that all patients had a single timepoint screen rather than repeated screens for AF? This point is pertinent for paroxysmal AF perhaps and that a single point screening is possibly an underestimate 

Abstract methods and findings- the last sentence should be a limitation of your study design. If the lack of OAC initiation is a reflection of the healthcare system in China, this should not be listed as a limitation but in the conclusion section

Abstract- you note that only 17 out of 103 patients attended speciality clinics. Can you clarify if these individuals were not referred to a speciality centre or if they did not attend of their own accord? This will impact how you frame the conclusions 

Line 2- is atrial fibrillation a growing problem? This implies that the incidence has increased. If this is true please provide a reference to support this and also reword for clarity 

Methods-could you name the participating hospitals here? Shanghai is a bit state, you may want to highlight the participating communities via a map

Did your study have a prospective protocol or analysis plan? Please state this (either way) early in the Methods section. a) If a prospective analysis plan (from your funding proposal, IRB or other ethics committee submission, study protocol, or other planning document written before analyzing the data) was used in designing the study, please include the relevant prospectively written document with your revised manuscript as a Supporting Information file to be published alongside your study, and cite it in the Methods section. A legend for this file should be included at the end of your manuscript. b) If no such document exists, please make sure that the Methods section transparently describes when analyses were planned, and when/why any data-driven changes to analyses took place. c) In either case, changes in the analysis-- including those made in response to peer review comments-- should be identified as such in the Methods section of the paper, with rationale. 

Please ensure that the study is reported according to the STROBE guideline, and include the completed STROBE checklist as Supporting Information. Please add the following statement, or similar, to the Methods: "This study is reported as per the Strengthening the Reporting of Observational Studies in Epidemiology (STROBE) guideline (S1 Checklist)." The STROBE guideline can be found here: http://www.equator-network.org/reporting-guidelines/strobe/ When completing the checklist, please use section and paragraph numbers, rather than page numbers.

Methods- please provide a copy of the medical history, lifestyle questionnaire used 

Methods- please do not use the phrase “elderly subjects” as 65 years and over constitutes a large age range 

Abstract, introduction and methods- it is unclear if the educational “intervention” should be renamed since this is not a trial? Please also clarify if the education and advice was given to patients directly or given to community health centre staff. 

Results- please provide details of how many patients at the community health centre were approached, how many declined to participate. 

Throughout- please use lower case p for p- values 

Throughout- where you mention ESC guidelines- please ensure you introduce ESC on first view 

Line 108-113 this information should be provided much sooner and more clearly. Please also provide additional details about the education provided at baseline. 

Conclusion- I would probably avoid the phrase "big gap" as this is subjective. 

PLOS Medicine requires that the de-identified data underlying the specific results in a published article be made available, without restrictions on access, in a public repository or as Supporting Information at the time of article publication, provided it is legal and ethical to do so. Please see the policy at 

http://journals.plos.org/plosmedicine/s/data-availability

and FAQs at 

http://journals.plos.org/plosmedicine/s/data-availability#loc-faqs-for-data-policy

Throughout- please provide 95% CI along with p-values

Comments from the reviewers:

Reviewer #1: Chen and colleagues used AliveCor to screen elderly Chinese patients for AF in Community Health Centers. They diagnosed newly detected AF in 0.5%, and actionable AF (either newly detected or undertreated known AF with high stroke risk) in 2.8%. Referrals were then made to specialty clinics for consideration of OAC. Disappointingly, not a single patient initiated OAC as a result of this screening program. This study is humbling and highlights the challenges to appropriate treatment with OAC for elderly Chinese patients. The authors appropriately discuss these numerous challenges.

The contribution of this study is incremental. As reviewed in the Discussion, many others have implemented more successful AF screening programs, resulting in effective initiation of OAC. The contribution here is mainly in highlighting the challenges to implementation of OAC in this specific population. 

One could question the role of ECG screening in patients with known AF. If a patient is known to have AF and has not initiated OAC, why would another AliveCor ECG documenting AF help?

Consider adding this citation to your Discussion: Heart Rhythm. 2019 Aug;16(8):e59-e65.

Reviewer #2: This is a statistical review of manuscript PMEDICINE-D-19-04467. The manuscript is very clear and well-written. I only have minor comments. 

Line 76: "For all analyses, a two-sided probability value < 0.05 was considered statistically significant." In the statistical jargon, it is actually referred to as a two-sided "p-value". 

Line 98: "OAC medication rate was 20% in all patients with known AF at baseline (28/138 ESC guideline recommended for OAC)". Can I just clarify that I understand the numbers correctly? There are 161 patients with known AF, and 183 patients with AF in total. 155/183 had CHA2DS2-VASc score >= threshold. Then out of the 161 with known AF, 138 were guideline recommended and only 28 were taking AOC? Figure 2 is excellent by the way and all answers are there, but perhaps you could you provide the breakdown of the 155 in the text so that it's easier when you read and don't have the Figure in front of you. 

Line 115: "Follow-up occurred in 126/155 patients at 12-months". Did you do the follow up only in the 155 patients who had CHA2DS2-VASc score >= threshold? If yes, could you please clarify in the Methods why this is the case? 

Line 119: there is a typo in the word "stroke" that is currently spelled as "stoke". 

Reviewer #3: Dear Authors

Thank you for preparing this manuscript. I enjoyed reading it.

You performed a single-arm study at community health centres the Shanghai area. You performed AF screening with a hand-held device. You found existing known AF in 3.5% of patients and new AF in 0.5% of patients. 2.8% of the overall population had AF that was not anti-coagulated. Over one year, a small proportion of patients attended specialist clinics and initiated OAC. You concluded that there was a significant gap between AF detection and treatment.

Your paper addresses and important issue. It is clear and easy to read. The design is generally appropriate. I think it is publishable. I do, however have a few comments that I believe would improve the study prior to publication. Most of these focus on the clarity of your methods and focusing on the findings of your study in the discussion.

I do think that this manuscript could benefit from English language editing. 

Abstract

Methods and findings

 You should be clear in the abstract that it was a single measurement for AF.

I don't agree with your statement that "The main limitation of our study is that the investigators were not able to provide patients who needed OAC therapy with either warfarin or non-warfarin oral anticoagulants .." This is not a limitation of your study, but a limitation of your healthcare system and should be framed as such.

Study population

 Can you please describe exhaustively and precisely how the program was advertised? Was it only by notice to the neighborhood committee and by placements of posters? If so, this sentence should read "The screening program was publicized through official notices to the neighborhood committee and placement of posters in community health centers." The current wording makes it sound like there were other channels

Screening and intervention

 Can you provide more description of the community health clinic? This would help with the generalizability and reproducibility off the study. Is it physician run? Are there other health professionals? Is this the usual place that study participants access primary care or is this more of a drop in?

 Rather than state that the ECGs were reviewed by an "investigator" or by a "the first author", this should be framed in the context of qualifications. Was it an arrhythmia specialist, cardiologist who reviewed them?

 Please provide a citation for OAC guidelines.

 Can you provide a more detailed description of the educational intervention? Was there a standard script? Did you use any patient decision-making tools? Was it done by a physician or other health care professional? How did you re-contact them at 1 month? It may be reasonable to provide a general overview in the methods and a detailed overview in an appendix.

 Can you describe the AF specialist clinics better? How many are there? Are they near the community health centres? Is the only place in the region where anticoagulants are prescribed? In most other countries, family physicians or generally practitioners are competent and comfortable with OAC prescription. 

Results

 The results section should begin with some broader statement of the eligible population. This is vital to understand the uptake of screening and the subset of the population that did participate. Can you provide some estimate of the number of potentially eligible persons in the cachement area of the clinic? Perhaps you have census data or some other estimate of the roster of patients at the community clinics?

The statement "Because warfarin, non-Vitamin K oral coagulants, and INR testing are not available at these community health centers, patients were advised to attend specialist clinics for OAC

prescription with information on how to make an appointment." Does not belong in the results. It should be in the methods, with further detail as per my comment above.

Discussion

I think your discussion is too long. I would aim to shorten it by at least 33%

It is interesting that among patients with actionable AF "only 17 attended cardiovascular specialist clinics and 4 of these were prescribed OAC" That is a discouragingly low rate and suggests that there is another critical barrier at the level of the specialist clinic. This should be re-inforced and discussed.

You postulate reasons why patients did not attend specialist clinics, but only one of these are supported directly by the findings of your study or by references. You also postulate possible interventions that could help improve OAC adherence. This paragraph has the same issues.

These two paragraphs could be combined, shortenend and amended. 

I don't think that the paragraph that begins "Our community study found significant under-treatment of known AF in sinus rhythm … is particularly useful. Given that you took only a single ECG and you don't have the power to do comparative statistics between the 3 groups, it is largely an over-interpretation of your data and it distracts from your overall message.

[LINK]

---

## [Decision Letter · Decision Letter 1]

30 Apr 2020

Dear Dr. Wang,

Thank you very much for re-submitting your manuscript "Detection rate and treatment gap for atrial fibrillation identified through screening and an education program in community health centers in China (AF-CATCH): a prospective cross-sectional study" (PMEDICINE-D-19-04467R1) for review by PLOS Medicine.

I have discussed the paper with my colleagues and the academic editor and it was also seen again by 3 reviewers. I am pleased to say that provided the remaining editorial and production issues are dealt with we are planning to accept the paper for publication in the journal.

[LINK]

We look forward to receiving the revised manuscript by May 07 2020 11:59PM. 

Sincerely,

Adya Misra, PhD

Senior Editor 

PLOS Medicine

plosmedicine.org

Requests from Editors:

Title-the title currently indicates the education program was carried out in community health centres. Please consider revising to “Detection rate and treatment gap for atrial fibrillation identified through screening in community health centers in China (AF-CATCH): a prospective cross-sectional study”

Abstract

Please add brief participant demographics 

Methods

INR testing? please define on first view

Informed consent-please mention details of consent within the methods section

Please also mention any additional follow-up after the 12 month follow-up. If patients were still not on OACs but have AF, were they referred to a specialist clinic again?

Table1 – p-values to three decimal places is sufficient 

The excel file containing de-identifying data contains data that may breach patient confidentiality such as date of birth. Please remove this information for privacy reasons and deposit these files with your local research ethics committee instead. The data availability statement should be revised to note that there are ethical restrictions on data sharing. 

Also, at this point please amend the statement in your methods section “All the records and information on participants were anonymized and de-identified before the analysis” as these do not appear to be de-identified medical records.

STROBE checklist- please check the provided checklist and corresponding paragraph numbers as they do not match the main text. For instance, you say example the discussion is in paragraph 7, but it's not. Please correct and clarify this. 

Comments from Reviewers:

Reviewer #1: While my concerns remain about the incremental contribution beyond previous screening studies and the lack of success in initiating anticoagulation, the authors have made an extensive effort to revise this manuscript and address all reviewer comments.

Reviewer #2: This is a statistical review of manuscript PMEDICINE-D-19-04467_R1. I thank the authors for their answers for my previous comments, which are satisfactory. I do not have further comments. 

Reviewer #3: Thank you for allowing me to review a revised version of your manuscript.

You have satisfactorily addressed all of the comments I raised in my first review and you have submitted a greatly improved paper.

I have one final comment that stems from a change you made between the original submission and first submission.

Your title describes the study as a prospective, cross-sectional study.

Prospective is correct, but cross-sectional is not.

Prospective refers to measuring an exposure at baseline (in your case AF) and then an outcome (OAC use) at some point in the future.

Cross-sectional would mean you measured them both at a single time point.

[LINK]

---

## [Editor Report · Decision Letter 2]

16 Jun 2020

Dear Prof. Wang, 

On behalf of my colleagues and the academic editor, Dr. William McIntyre, I am delighted to inform you that your manuscript entitled "Detection rate and treatment gap for atrial fibrillation identified through screening in community health centers in China (AF-CATCH): a prospective multi-center study" (PMEDICINE-D-19-04467R2) has been accepted for publication in PLOS Medicine. 

PRODUCTION PROCESS

PRESS

PROFILE INFORMATION

Thank you again for submitting the manuscript to PLOS Medicine. We look forward to publishing it. 

Best wishes, 

Adya Misra, PhD

Senior Editor 

PLOS Medicine

plosmedicine.org